# Comparative Study of the SEM Evaluation, EDX Assessment, Morphometric Analysis, and Cyclic Fatigue Resistance of Three Novel Brands of NiTi Alloy Endodontic Files

**DOI:** 10.3390/ijerph19074414

**Published:** 2022-04-06

**Authors:** Vicente Faus-Matoses, Raúl Pérez García, Vicente Faus-Llácer, Ignacio Faus-Matoses, Óscar Alonso Ezpeleta, Alberto Albaladejo Martínez, Álvaro Zubizarreta-Macho

**Affiliations:** 1Department of Stomatology, Faculty of Medicine and Dentistry, University of Valencia, 46010 Valencia, Spain; vicente.faus@uv.es (V.F.-M.); raulpe8@alumni.uv.es (R.P.G.); fausvj@uv.es (V.F.-L.); 2Department of Endodontics, School of Health Sciences, University of Zaragoza, 22006 Zaragoza, Spain; lalonezp@unizar.es; 3Department of Surgery, Faculty of Medicine and Dentistry, University of Salamanca, 37008 Salamanca, Spain; albertoalbaladejo@hotmail.com (A.A.M.); or amacho@uax.es (Á.Z.-M.); 4Department of Endodontics, Faculty of Health Sciences, Alfonso X El Sabio University, 28691 Madrid, Spain

**Keywords:** cyclic fatigue, endodontics, energy-dispersive X-ray, scanning electron microscopy, morphometry, reciprocating, continuous rotation, NiTi

## Abstract

In this study, we compare and analyze the scanning electron microscopy (SEM), energy-dispersive X-ray spectroscopy (EDX), morphometry and cyclic fatigue resistance of Endogal, PathMax, and Smarttrack novel brands of nickel–titanium (NiTi) alloy endodontic files. Material and Methods: Thirty sterile NiTi endodontic rotary files were randomly selected and assigned to one of the following study groups: A: 25.08 F2 Endogal (EDG) (*n* = 10); B: 25.08 F2 Path Max Pro (PMP) (*n* = 10); and C: 25.06 Smarttrack (ST) (*n* = 10). Dynamic cyclic fatigue tests were conducted using a cyclic fatigue device in stainless steel artificial root canal systems with an apical diameter of 250 µm, curvature angle of 60°, radius of curvature of 3 mm, and taper of 6%. Additionally, we analyzed the NiTi endodontic rotary files using EDX, SEM, and morphometry after micro-CT scanning. The results were analyzed using Weibull statistical analysis and ANOVA testing. Results: SEM, EDX, and morphometric analyses showed differences between the three novel brands of NiTi endodontic rotary files. Moreover, statistically significant differences were observed between the number of cycles to failure and time to failure of the three novel brands of NiTi endodontic rotary files (*p* < 0.001). **Conclusions**: Smarttrack NiTi alloy endodontic reciprocating files display greater resistance to cyclic fatigue than Endogal and Path Max Pro NiTi alloy endodontic rotary files, due to the reciprocating movement and metallurgical composition.

## 1. Introduction

The use of nickel–titanium (NiTi) alloy for the manufacturing of instruments used in root canals has revolutionized the field of endodontics, with these new endodontic instruments reducing iatrogenic complications [1,2]. Nevertheless, failure of these instruments is still a concern for clinicians. Despite the continued chemical and mechanical improvements to the NiTi alloy endodontic rotary instruments by their manufacturers in an effort to prevent potential root canal treatment complications [3], fractures of NiTi endodontic rotary files still occur, with an incidence rate between 0.09% and 5% [4,5]. These nickel–titanium alloy endodontic rotary files fail when their resistance to fatigue is surpassed by flexural bending (cyclic) stress, torsional stress or a combination thereof [6]. More precisely, torsional fatigue is seen when the tip of the endodontic file becomes blocked within the root canal as the instrument still rotates [7]. Flexural bending fatigue occurs when alternating tensile and compressive stress cycles applied on a curved root canal result in plastic deformities and the endodontic rotary instrument subsequently fails [7,8], despite the centering ability and contact pressure distribution of the NiTi endodontic rotary file against the root canal walls during shaping of the root canal system [9,10].

Furthermore, unexpected failures of NiTi alloy endodontic rotary instruments can inhibit good clinical outcomes by blocking disinfecting agents from penetrating past the fractured instrument [11,12,13], potentially leading to pulp necrosis and causing periapical lesions to form [14]. This can also affect the success rates of root canal treatments in teeth with periapical pathologies [15]. In an effort to address these issues, several studies have assessed the effect of both nickel–titanium alloys and geometric parameters on the resistance to flexural bending and torsional fatigue of endodontic rotary instruments, with the aim of lowering the incidence of failure in these instruments. The crystalline structure and chemical make-up of the nickel–titanium alloy have both been extensively analyzed as factors affecting the fatigue resistance of endodontic rotary files, especially in endodontic rotary systems with a more highly concentrated martensitic phase or those manufactured by electro-polishing, ion implantation, cryogenic treatment, and heat treatment to improve the mechanical behaviors of NiTi endodontic rotary files; particularly, increasing their resistance to cyclic fatigue [16]. Nevertheless, researchers have reported that certain geometric factors also influence instrument performance, including the apical diameter and taper [17], flute length, cross-section design [18,19], pitch [18], brushing movement [20], and helix angle. In addition, Nóvoa et al. evidenced the corrosive effect of a sodium hypochlorite irrigating solution on NiTi alloy endodontic rotary files [21]. However, Cavalleri et al. reported that this corrosive effect does not increase the risk of fracture of the NiTi alloy endodontic rotary instrument [22]. Unfortunately, it is difficult to independently assess each factor that is linked to flexural bending fatigue within a clinical setting, due to the heterogeneous anatomical make-up of root canal systems. As a result, controlled experimental studies have enabled the independent analysis of each variable using custom-made cyclic fatigue devices.

More recently, novel brands of NiTi alloy endodontic instruments, such as Endogal, PathMax, and Smarttrack, have emerged to enhance the mechanical properties and reduce the rates of failure of current instrumentation systems [23]. However, the absence of literature related to the geometrical design, metallurgical composition, and mechanical properties require a study to analyze the metallurgical composition and compare the cyclic fatigue resistance of these instruments.

In the present study, we aim to analyze and compare the scanning electron microscopy, energy-dispersive X-ray spectroscopy, morphometry and cyclic fatigue resistance of Endogal, PathMax, and Smarttrack novel brands of NiTi alloy endodontic files, with a null hypothesis (H_0_) stating that Endogal, PathMax, and Smarttrack novel brands of NiTi alloy endodontic files do not show differences in the scanning electron microscopy, energy-dispersive X-ray spectroscopy, morphometry and cyclic fatigue resistance.

## 2. Materials and Methods

### 2.1. Study Design

Thirty unused NiTi alloy endodontic instruments were utilized for this in vitro study. The endodontic rotary files were supplied as sterile by the manufacturer. A total of 30 experimental units were included, distributed among the 3 groups, in accordance with the proportions set by the researcher, and a power of 80% was set. In addition, an effect size of 0.606 can be detected when testing the null hypothesis H₀: The means of the 3 groups were equal by the means of a 1-factor ANOVA test for independent samples, taking into account that the level of significance is 5%. Between March and July 2021, a controlled experimental study was carried out at the Department of Stomatology of the Faculty of Medicine and Dentistry at the University of Valencia (Valencia, Spain). NiTi alloy endodontic rotary files were randomly selected and distributed among the following study groups: A: 250 µm apical diameter and 6% taper gold-wire NiTi alloy endodontic rotary file (Ref.: IRE 02506, D, Endogal, Galician Endodontics Company, Lugo, Spain) (*n* = 10) (EDG); B: 250 µm apical diameter and 6% taper heat-treated NiTi alloy endodontic rotary file (Ref.: 121 812, PathMax Pro, Nikinc Dental B.V., Eindhoven, The Netherlands) (*n* = 10) (PMP), and C: 250 µm apical diameter and 6% taper heat-treated NiTi alloy endodontic reciprocating file (Ref.: 121 104, 25, Smarttrack, Nikinc Dental B.V., Eindhoven, The Netherlands) (*n* = 10) (ST).

### 2.2. Scanning Electron Microscopy Analysis

All of the NiTi alloy endodontic files were initially analyzed using a scanning electron microscope (SEM) (HITACHI S-4800, Fukuoka, Japan) at 30× and 600× magnification. This analysis was carried out at the Central Support Service for Experimental Research of the University of Valencia in Burjassot, Spain, and was conducted using the following exposure parameters: 20 kV acceleration voltage, resolution ranging from − 1.0 nm at 15 kV to 2.0 nm at 1 kV, and magnification from 100× to 6500×. This was carried out for surface characterization and to rule out further surface defects due to manufacturing, as well as to compare and contrast the geometric designs of the Endogal, PathMax, and Smarttrack novel brands of NiTi alloy endodontic files.

### 2.3. Energy-Dispersive X-ray Spectroscopy Analysis

In addition, an energy-dispersive X-ray spectroscopy (EDX) analysis was carried out on all of the NiTi files. This was conducted at the Central Support Service for Experimental Research of the University of Valencia in Burjassot, Spain, and the following exposure parameters were used: Acceleration voltage of 20 kV, resolution ranging from −1.0 nm at 15 kV to 2.0 nm at 1 kV, and magnification from 100× to 6500×. This analysis was carried out to assess the elemental make-up of the chemical elements of the files. Moreover, we assessed atomic weight percentages, with measurements taken at three different locations.

### 2.4. Micro-Computed Tomography Scan and Morphometric Analyses

Finally, a micro-CT scan (Skyscan 1176, Bruker-MicroCT, Kontich, Belgium) was performed to obtain accurate Digital Imaging and Communications in Medicine (DICOM) digital files of the NiTi alloy endodontic files, at the Department of Mechanical, Energetic, and Materials Engineering of the School of Industrial Engineering at the University of Valencia (Valencia, Spain), under the following exposure parameters: 56.0–58.0 microamperes, 160.0 kilovolt peak, 500.0 ms, 4 frames, 720 projections, 3 µm resolution, a tungsten target from 0.25–0.375 mm, and pixel size of 0.127 µm. Thereafter, morphometric analysis of the pitch and helix angle of the NiTi endodontic rotary files was carried out using Fiji/ImageJ (Oviedo, Spain), an open-source Java-based image processing software [24]. Subsequently, each DICOM digital file was segmented from the micro-CT volume to quantify its pitch and helix angle, by thresholding the original volume to obtain a 3D binary mask. Then, the longitudinal axis of the NiTi alloy endodontic files was established by interpolating the center of each Z-slice (file section). Thereafter, the maximum (peak) and minimum (valley) distances were defined in a longitudinal view between the file edge and the longitudinal axis to define the different pitches. Finally, the direction of the cutting edge over the longitudinal axis was overlapped to determine the helix angle. The pitch and helix angle measurements was recorded at all of the pitches from the file tip.

In addition, the DICOM digital files were transferred to Standard Tessellation Language (STL) digital files for the Endogal, PathMax, and Smarttrack novel brands of NiTi alloy endodontic files to illustrate the values of the pitch and helix angle, as well as to contrast the geometric designs of the files.

Furthermore, the cross-section geometry of Endogal, PathMax, and Smarttrack novel brands of NiTi alloy endodontic files was analyzed.

### 2.5. Experimental Cyclic Fatigue Model

We conducted dynamic cyclic fatigue resistance tests using a custom-made device (Utility Model Patent No. ES1219520) [25]. The structure of the device was designed using CAD/CAE 2D/3D software (Midas FX+^®^, Brunleys, Milton Keynes, UK), and printed using the 3D printing software (ProJet^®^ 6000 3D Systems^©^, Rock Hill, SC, USA); see Figure 1.

The custom-made artificial root canals were designed with a 60° curvature, as per Schneider’s measuring technique [26], with a radius of curvature of 3 mm. For this, the CAD/CAE 2D/3D inverse engineering software was used. The artificial root canal was manufactured from stainless steel with electrical discharge machining (EDM) molybdenum wire-cut technology (Cocchiola S.A., Buenos Aires, Argentina), thereby ensuring close contact between the NiTi files and the walls of the artificial root canal (Figure 2).

Then, the manufactured canal was positioned on its support, and any failures in the endodontic rotary instruments were detected using a light-dependent resistor (LDR) sensor (Ref.: C000025, Arduino LLC^®^, Ivrea, Italy) positioned at the apex of the canal. The LDR sensor measures the continuous light source of a very bright white LED (20000 mcd) (Ref.: 12.675/5/b/c/20 k, Batuled, Coslada, Spain). This LED was placed opposite to the artificial root canal (Figure 1). The LED signals were picked up by the LDR sensor (Ref.: C000025, Arduino LLC^®^) every 50 ms to pinpoint the exact time of failure.

The speed and movement direction were controlled by a driver (Ref.: DRV8835, Pololu^®^ Corporation, Las Vegas, NV, USA), generated by a brushed DC gear motor (Ref.: 1589, Pololu^®^ Corporation, Las Vegas, NV, USA), and applied to the artificial support using a roller bearing system (Ref.: MR104ZZ, FAG, Schaeffler Herzogenaurach, Germany). The support moved in a solely axial motion using a lineal guide (Ref.: HGH35C 10249-1 001 MA, HIWIN Technologies Corp. Taichung, Taiwan). A 6:1 reduction handpiece (X-Smart plus, Dentsply Maillefer, Baillagues, Switzerland) and torque-controlled motor were used, in accordance with the manufacturer’s instructions (Table 1).

All of the NiTi alloy endodontic files were used at a frequency of 60 pecks per minute within the dynamic cyclic fatigue device, as per a prior study. To prevent friction between the reciprocating files and the artificial root canal walls, we applied a specialized high-flow synthetic oil (Singer All-Purpose Oil; Singer Corp., Barcelona, Spain) formulated for use in lubricating mechanical parts.

All of the files were used until they fractured. We measured and recorded the time to failure and the number of cycles to failure.

### 2.6. Statistical Tests

All of the variables were statistically analyzed using the SAS 9.4 software (SAS Institute Inc., Cary, NC, USA). The descriptive statistics of quantitative variables were expressed as mean and SD. Then, we performed a comparative analysis, using the ANOVA test to compare the time to failure (in seconds) and number of cycles to failure. In 2-to-2 comparisons, Tukey’s test was used to adjust the *p*-values to correct for Type I error. Furthermore, the Weibull modulus and characteristic strength were calculated. The statistical significance was defined as *p* ˂ 0.05.

## 3. Results

SEM analysis of the Endogal, PathMax, and Smarttrack novel brands of NiTi alloy endodontic files showed accumulated organic matter, but no structural alterations were observed in any of the files. Manufacturing lines were perpendicularly distributed, in relation to the longitudinal axis, in all of the files. They were also parallel, as a result of manufacturing with laser machining. The tubular porosity as well as the width and spacing of the manufacturing lines both correspond to the intensity and precision of the laser machining process. Furthermore, the microscopic geometric design of EDG NiTi alloy endodontic rotary files, PMP NiTi alloy endodontic rotary files, and ST NiTi alloy endodontic reciprocating files clearly differed in the tip design; particularly, in the direction of the helix angle of ST NiTi alloy endodontic reciprocating files (Figure 3).

EDX micro-analysis of EDG, PMP, and ST NiTi alloy endodontic files was carried out at 15 and 20 kV. However, the 20 kV acceleration voltage results were more determinant than those at 15 kV. In summary, the EDG, PMP, and ST NiTi alloy endodontic files differed in atomic weight percentages, although the elements present in the chemical composition were similar (Figure 4 and Table 2).

The morphometric analysis showed a lower pitch of the EDG NiTi alloy endodontic rotary file (6 pitches), compared with the PMP heat-treated NiTi alloy endodontic rotary file (10 pitches) and ST heat-treated NiTi alloy endodontic reciprocating file (10 pitches) (Figure 5). In addition, the EDG NiTi alloy endodontic rotary file showed a more pronounced gradual reduction of pitch to the tip, followed by the PMP heat-treated NiTi alloy endodontic rotary file and ST heat-treated NiTi alloy endodontic reciprocating file (Figure 5). Moreover, the replica-like EDG NiTi endodontic rotary files did not show measuring lines in the non-cutting part (Figure 5). Furthermore, the morphometric analysis showed different cross-sectional geometries between the novel brands of NiTi alloy endodontic files; particularly, between the EDG NiTi alloy endodontic rotary files and PMP heat-treated NiTi alloy endodontic rotary file and ST heat-treated NiTi alloy endodontic reciprocating file (Figure 6).

In addition, EDG NiTi endodontic rotary files showed a gradual reduction of the helix angle from the tip. However, ST heat-treated NiTi alloy endodontic reciprocating files showed a slight gradual reduction of the helix angle and PMP heat-treated NiTi alloy endodontic rotary files did not show variations of the helix angle along the cutting surface, which is a similar helix angle along the cutting part of the files (Table 3). Furthermore, ST heat-treated NiTi alloy endodontic reciprocating files showed a different direction of the helix angle, compared with EDG NiTi endodontic rotary files and PMP heat-treated NiTi alloy endodontic rotary files (Figure 5).

The mean and SD values of time to failure (in seconds) for each of the study groups can be seen in Table 4.

The ANOVA test revealed statistically significant differences between the mean time to failure for all novel brands of NiTi alloy endodontic files (*p* ˂ 0.001). In addition, Tukey’s test indicated statistically significant differences between NiTi endodontic rotary files and NiTi endodontic reciprocating files (*p* ˂ 0.001). The results for number of cycles to failure were similar, as the dynamic cyclic fatigue device was set at a frequency of 60 pecks per minute.

The Weibull statistic scale distribution parameter (η) indicated statistically significant differences in time to failure across all of the novel brands of NiTi alloy endodontic files (*p* ˂ 0.001; Table 5, Figure 7). On the other hand, the shape distribution parameter (β) revealed no statistically significant differences in time to failure for any of the novel brands of NiTi alloy endodontic files (*p* ˃ 0.05). The results for number of cycles to failure were similar, as the dynamic cyclic fatigue device was set at a frequency of 60 pecks per minute (Table 5, Figure 7).

## 4. Discussion

Our results reject the null hypothesis (H_0_) and Tukey’s test also found that Endogal, PathMax, and Smarttrack novel brands of NiTi alloy endodontic files do not show differences in the scanning electron microscopy, energy-dispersive X-ray spectroscopy, morphometry and cyclic fatigue resistance.

The results of the SEM analysis indicated that the microscopic geometric design of the EDG NiTi alloy endodontic rotary files, PMP heat-treated NiTi alloy endodontic rotary files, and ST heat-treated NiTi alloy endodontic reciprocating files were clearly different. Furthermore, EDX micro-analysis showed that the atomic weight percentages were different in the atomic weight percentage, although the elements present in the chemical composition were similar. The largest difference in chemical composition was shown in the EDG NiTi alloy endodontic rotary file, since it incorporated a high concentration of C in the chemical make-up of the NiTi alloy (27.66–45.60 wt%). Titanium is an allotropic metal with compact hexagonal (α or austenite) or body-centered cubic (β or martensite) structures. Depending on the stabilizing effects of the α and β phases, titanium’s alloying elements are defined as neutral elements (i.e., stabilizers of the α phase or betagenic elements) or as alphagenic elements (i.e., stabilizers of the β phase). Phase stabilization signifies a higher or lower transition temperature β [27]. More specifically, alphagenic elements increase the transition temperature β. Of the alphagenic elements, Al is the most significant alloying element, although C, O, and N can also be used. The results of the EDX micro-analysis revealed the presence of C and O alloying elements, leading to a more martensitic crystalline structure. Therefore, this makes them more flexible and fracture-resistant. A higher atomic weight percentage of oxygen was shown after the EDX micro-analysis of PMP heat-treated NiTi alloy endodontic rotary files (38.79–39.78 wt%), compared with EDG NiTi alloy endodontic rotary files (24.05–25.41 wt%) and ST heat-treated NiTi alloy endodontic reciprocating files (24.79–27.57 wt%), which would lead to the higher resistance to cyclic fatigue of the PMP heat-treated NiTi alloy endodontic rotary files.

Furthermore, morphometric analysis of the novel brands of NiTi alloy endodontic files showed differences at the pitch and helix angle and the cross-section geometric design; particularly between the EDG NiTi alloy endodontic rotary file (trapezoidal) and the PMP heat-treated NiTi alloy endodontic rotary file (convex triangular) and ST heat-treated NiTi alloy endodontic rotary file (convex triangular). PMP heat-treated NiTi alloy endodontic rotary file and ST heat-treated NiTi alloy endodontic rotary file showed a higher number of threads than the EDG NiTi alloy endodontic rotary file. Moreover, Versluis et al. experimentally reported that the number of threads directly increases the flexural stiffness [28]. However, Al Raeesi et al. found that a shorter pitch design increased the resistance to cyclic fatigue of glide path instruments [29]. The EDG NiTi alloy endodontic rotary file and ST heat-treated NiTi alloy endodontic rotary file showed a gradual trend of reducing the helix angle to the tip. However, the PMP heat-treated NiTi alloy endodontic rotary file maintained a continuous helix angle along the cutting part of the endodontic rotary file. In addition, Rui et al. experimentally demonstrated that an increase in the helix angle value improves the mechanical behavior of the instrument under torsional and bending conditions [30].

Furthermore, Faus-Llacer et al. found that two NiTi-alloy endodontic rotary files with a double S-shaped cross-section were more resistant to dynamic cyclic fatigue than T Pro E1 austenite phase nickel–titanium alloy endodontic rotary files with a rectangular cross-section, T Pro E2 austenite phase NiTi alloy endodontic rotary files with a convex triangular cross-section, and T Pro E4 austenite phase NiTi alloy endodontic rotary files with a triangular cross-section [31]. These results indicate that, by increasing the mass and contact points between the instrument surface and dentin walls of the root canal, the resistance to cyclic fatigue of the NiTi alloy endodontic rotary files decrease. The flexibility of NiTi alloy endodontic rotary file may also be affected, resulting in excessive root canal dentine removal, root perforations, apical transport [32], and fractures [4,33,34]. Briefly, the results derived from the cyclic fatigue analysis demonstrated that the ST novel brand NiTi alloy endodontic reciprocating files had greater resistance to cyclic fatigue than the EDG and PMP novel brands of NiTi alloy endodontic rotary files.

The anatomical-based root canal design used in this study was based on Schneider’s method [19], using a radius of 3 mm and curvature angle of 60°. Herein, the geometry was adapted to the aforementioned NiTi endodontic rotary files. Prior studies have shown that the resistance to fatigue of files becomes lower as the radius of curvature decreases and the angle of curvature increases [12,35,36], as the stress accumulating on the file is inversely proportional to the radius of the canal’s curvature. Consequently, more abrupt root canals have greater torsion and flexural bending fatigue, which may ultimately lead to instrument fracture [17]. Furthermore, clinical or even ex vivo experimental studies are recommended, to better replicate clinical conditions and to make it possible to apply these cyclic fatigue results to a clinical setting. However, difficulties in homogenizing the curvature angle, radius, apical diameter, cross-section, and hardness of the root canals may bias the study results by adding more variables. Therefore, custom-made dynamic and static cyclic fatigue devices have been developed to independently analyze the effects of the variables under study. Unfortunately, at present, there are no norms regulating the characteristics of these custom-built devices, nor are there any international standards for testing the cyclic fatigue behaviors of NiTi alloy endodontic rotary instruments with a taper larger than 2% [37]. The development of additive manufacturing processes based on three-dimensional printing has allowed for the acquisition of anatomical-based dental replicas, which can be used for the analysis of the cyclic fatigue resistance of NiTi alloy endodontic rotary files [38]. However, the physical properties of the printed plastic material used in three-dimensional printing highly differ from those of the root dentin. Therefore, translation of the obtained results to clinical settings is difficult.

Cyclic fatigue has been studied using both static and dynamic testing. In static cyclic fatigue testing models, NiTi alloy endodontic files are rotated until they fracture, with tension–compression cycles concentrated in the maximum curvature angle of the root canal, causing microstructural changes to the file, and triggering its subsequent failure. Therefore, dynamic cyclic fatigue testing devices are preferred, as they can better simulate clinical conditions; particularly, the pecking motion of nickel–titanium endodontic rotary files. As a result, in this study, we employed an anatomically accurate artificial root canal, dynamic cyclic fatigue testing, and an automatic detection system to accurately and objectively identify failure of the endodontic rotary files [39,40].

Unfortunately, limitations to the present study prevented the NiTi alloy, pitch, helix angle, speed or manufacturing process from standardization. Furthermore, this study was not carried out in a clinical setting, as the associated samples are difficult to standardize.

## 5. Conclusions

The results obtained in the present study led us to conclude that: Smarttrack novel brand of NiTi alloy endodontic reciprocating files display greater resistance to cyclic fatigue than Endogal and Path Max Pro novel brands of NiTi alloy endodontic rotary files, due to the reciprocating movement and metallurgical composition.

## Figures and Tables

**Figure 1 ijerph-19-04414-f001:**
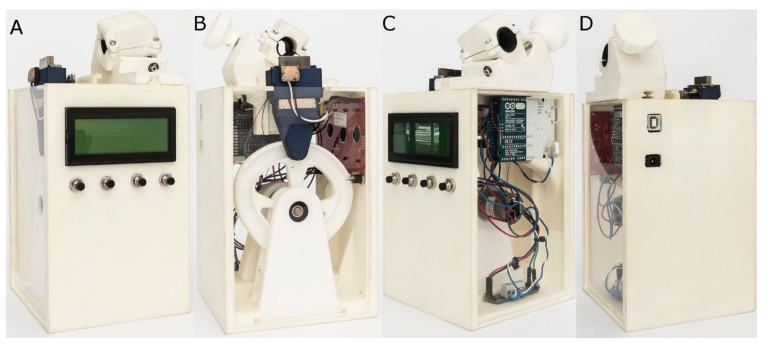
(**A**) Front, (**B**) back, (**C**) right side, and (**D**) left side of the dynamic cyclic fatigue device.

**Figure 2 ijerph-19-04414-f002:**
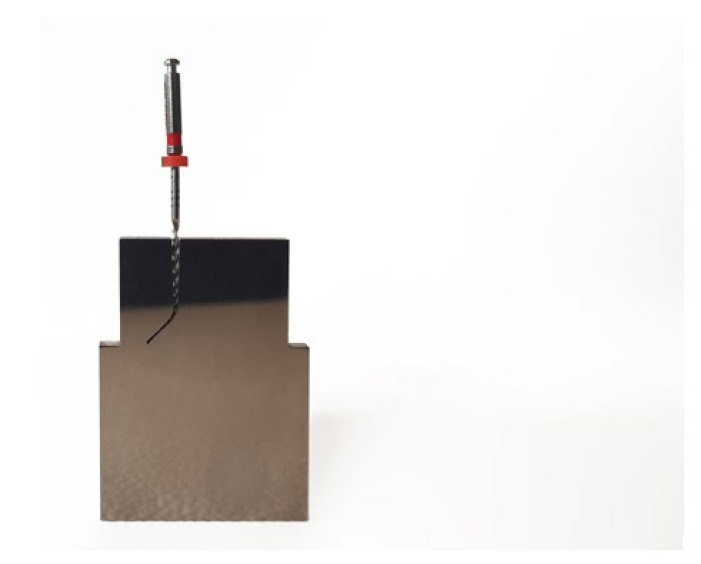
Artificial root canal manufactured from stainless steel with EDM molybdenum wire-cut technology in close contact with a NiTi alloy endodontic rotary file.

**Figure 3 ijerph-19-04414-f003:**
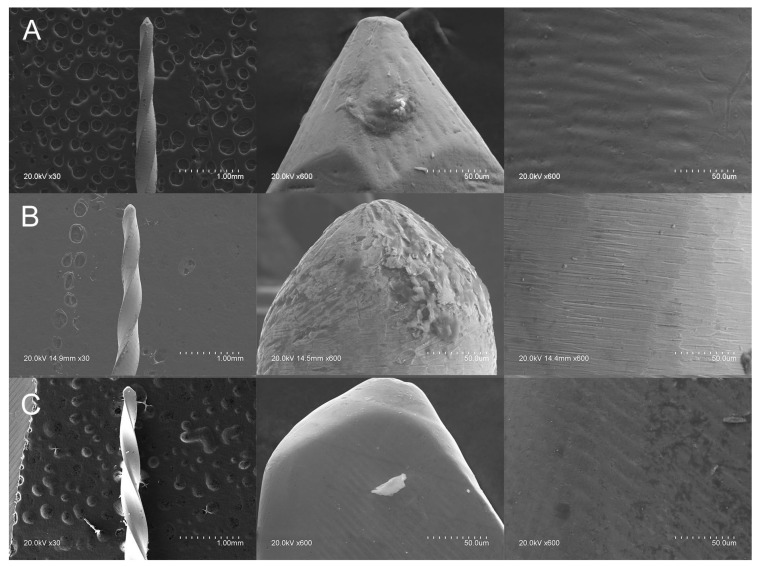
SEM analysis (at 30× and 600×) of: (**A**) EDG NiTi alloy endodontic rotary file, (**B**) PMP heat-treated NiTi alloy endodontic rotary file, and (**C**) ST heat-treated NiTi alloy endodontic reciprocating file.

**Figure 4 ijerph-19-04414-f004:**
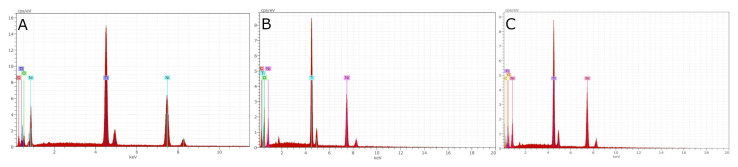
EDX micro-analysis of: (**A**) EDG NiTi alloy endodontic rotary file, (**B**) PMP heat-treated NiTi alloy endodontic rotary file, and (**C**) ST heat-treated NiTi alloy endodontic reciprocating file.

**Figure 5 ijerph-19-04414-f005:**
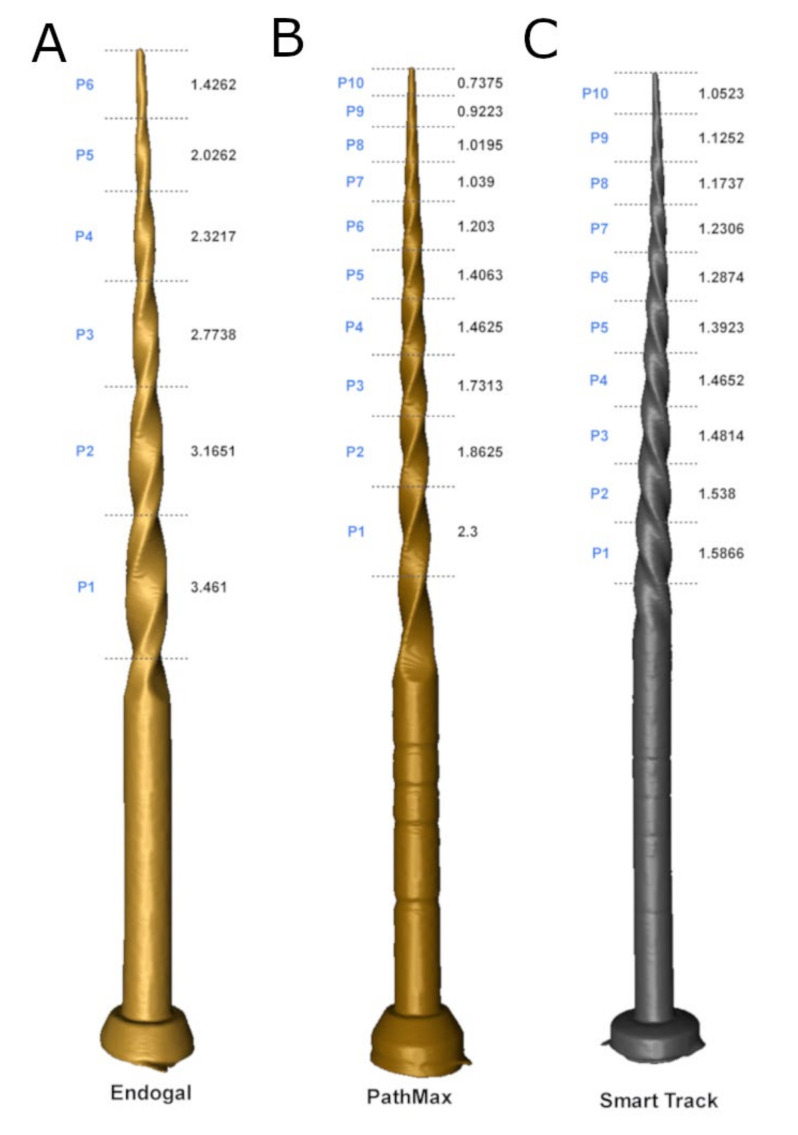
(**A**) STL digital files of EDG NiTi alloy endodontic rotary file, (**B**) PMP heat-treated NiTi alloy endodontic rotary file, and (**C**) ST heat-treated NiTi alloy endodontic reciprocating file.

**Figure 6 ijerph-19-04414-f006:**
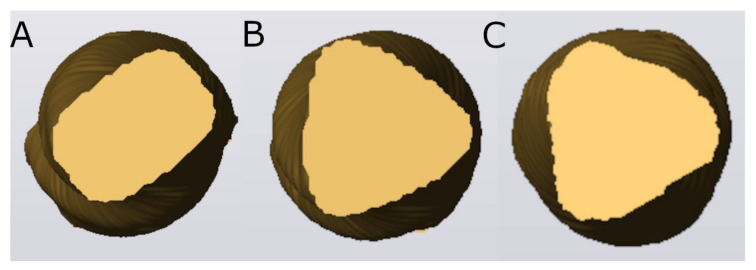
STL digital files of the cross-section geometries of: (**A**) EDG NiTi alloy endodontic rotary file, (**B**) PMP heat-treated NiTi alloy endodontic rotary file, and (**C**) ST heat-treated NiTi alloy endodontic reciprocating file.

**Figure 7 ijerph-19-04414-f007:**
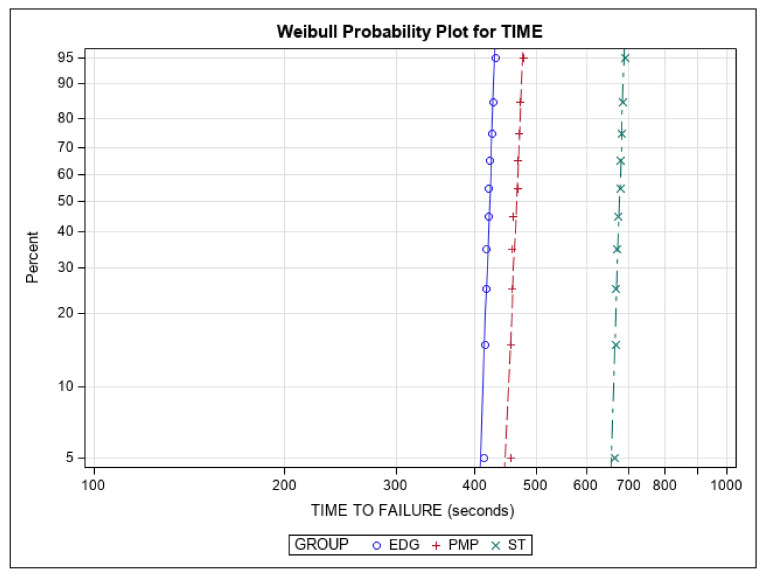
Weibull probability plot of time to failure for the EDG, PMP, and ST novel brands of NiTi alloy endodontic files.

**Table 1 ijerph-19-04414-t001:** Characteristics of the movement performed by each study group, in terms of speed, torque/direction of movement, and type of movement.

Study Group	Speed	Torque/Direction	Movement
EDG	250 rpm	4 N/cm	Continuous rotational
PMP	250 rpm	4 N/cm	Continuous rotational
ST	350 rpm	170° CCW/50° CW	Reciprocant

CCW: Counterclockwise; CW: Clockwise.

**Table 2 ijerph-19-04414-t002:** Mean atomic weight percentages (%) of EDG NiTi alloy endodontic rotary file, PMP heat-treated NiTi alloy endodontic rotary file, and ST heat-treated NiTi alloy endodontic reciprocating file at 15 and 20 kV in three different locations (1–3).

Spectrum	C	O	Ti	Ni
EDG 15 kV (1–3)	27.66	24.05	25.98	22.31
PMP 15 kV (1–3)	28.79	39.78	17.32	14.10
ST 15 kV (1–3)	29.57	27.57	22.69	20.18
EDG 20 kV (1–3)	45.60	25.41	16.13	12.86
PMP 20 kV (1–3)	30.74	38.79	16.44	14.03
ST 20 kV (1–3)	30.64	24.79	22.45	22.11

**Table 3 ijerph-19-04414-t003:** Mean helix angle measurements (°) at all of the pitches to the tip of novel brands of NiTi alloy endodontic files.

Study Group	P1	P2	P3	P4	P5	P6	P7	P8	P9
EDG	26.7	23.3	21.9	18.3	15.1	13.6			
PMP	25.5	25.4	26	24.9	25.7	26.3	26	26.6	25.2
ST	35.8	33.7	32.7	31.2	28	25.5	22.3	20.2	17.1

**Table 4 ijerph-19-04414-t004:** Descriptive statistics regarding time to failure of the novel brands of NiTi alloy endodontic files.

	*n*	Mean	SD	Minimum	Maximum
EDG	10	420.78 ^a^	5.83	413.00	431.10
PMP	10	463.97 ^b^	7.55	455.90	477.50
ST	10	675.95 ^c^	8.07	664.70	689.60

^a,b,c^ Statistically significant differences between groups (*p* < 0.05).

**Table 5 ijerph-19-04414-t005:** Weibull statistics of time to failure for the EDG, PMP, and ST novel brands of NiTi alloy endodontic files.

	Weibull Shape (β)				Weibull Scale (η)			
Study Group	Estimate	Std. Error	Lower	Upper	Estimate	Std. Error	Lower	Upper
EDG	76.9	18.0	48.5	121.9	423.5	1.8	419.9	427.2
PMP	65.6	15.4	41.4	104.1	467.6	2.3	462.9	472.3
ST	91.1	21.6	57.2	145.2	679.8	2.5	674.9	684.7

## Data Availability

Data available upon request, in accordance with relevant considerations (e.g., ethical privacy restrictions).

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
