# Peer review of "Comparative Study of the SEM Evaluation, EDX Assessment, Morphometric Analysis, and Cyclic Fatigue Resistance of Three Novel Brands of NiTi Alloy Endodontic Files"

_ijerph, 2022, doi:10.3390/ijerph19074414_

Round 1

Reviewer 1 Report

This study aims to compare and analyze the cyclic fatigue resistance of replica-like and original-brand endodontic rotary files made from nickel-titanium (NiTi).

This article has much merit, however, some serious methodological errors persist and make the study results invalid, preventing its publication. Moreover, the authors have not replied to almost any of the questions put to them. The manuscript still contains serious errors.

This study should only be published as a descriptive analysis of the 3 new file systems, but without comparing them with the previously existing systems on the market. No reference can be made to the fact that they are replicas. That is a lie. None of the new systems has the same alloy, the same taper, the same section, the same pitch, the same helicoidal angle, the same cutting angle.... as the supposed replica! Figures 2 and 3 itself show that the new brands and the supposed replicas are nothing alike. Figure 5 also demonstrates that the section of ST heat-treated NiTi alloy endodontic reciprocating file is very different to original-brand WOG Gold-wire.

Moreover, the authors themselves recognize in their answers that “From the present study we can affirm that the compared systems are not replicas, but different systems since differ metallurgically and geometrically; however, this was not known prior”.

1) Authors state “In fact, the manufacturers of Endogal, PathMax, and Smarttrack have reported that the geometric design and chemical composition of their instruments are similar and based on those of the ProTaper Next, ProTaper Gold, and Wave One Gold NiTi endodontic rotary instruments, respectively”. However, they do not provide any reference that proves that the manufacturers recognize the copy of other brands. Authors should cite references of manufacturers stating this. It is essential to prove that the manufacturers propose the new systems as copies of the previous ones, which could even bring them legal problems. The title should be modified.

2) In reference to the calculation of the sample size, authors state “We carried out an ANOVA to establish the sample size, achieving 80% power with a confidence level of 5%, with a variability between groups of 0.6 and intra-group variability of 4, in order to identify differences in the contrast of the null hypothesis H₀: μ1 = μ2 102 = μ3 = μ4”. However, they do not specify how many differences they intended to find between the samples. What program have they used? What has been the minimum sample size calculated? With the parameters they give, they would have needed more than 20 files in each group!  The sample is insufficient to validate the results.

3) One of the key questions I posed to the authors in my first review was the following:

Authors states “EDG NiTi endodontic rotary files (replica) were compared to PTN NiTi endodontic rotary files (original brand)”. Why do the authors assume that the 25 Endogal file is a replica of the 25 Protaper Next file? Please, include a reference to support this claim! A reference is also necessary to support that Endogal is a Gold NiTi alloy!, How can you compare the cyclic fatigue of a supposedly Gold martensitic alloy file with an austenitic M-Wire file like Protaper Next? Obviously, the result is already known! And the same happens with the rest of the supposed replicas.

In their answer, authors state: In order to adapt to the reviewer's comments, we clarify that the manufacturer of the replica-like Endogal NiTi endodontic rotary files ensures that they are replicas of the original-brand Protaper Next NiTi endodontic rotary files. But they do not provide any reference that proves that the manufacturers recognize the copy of other brand. Moreover, the Protaper Next section is completely different from the supposed replica Endogal. Files of different design, different NiTi alloy... cannot be compared to conclude that one has more resistance to cyclic fatigue than the other. It seems that the authors´ objective is to highlight that the replica is better than the original!

4) Authors state: “In addition, the replica-like Endogal NiTi alloy endodontic rotary files showed a significantly higher time…”. Obviously, since they are comparing a martensitic file (Endogal) with an austenitic file (Protaper Next). I insist, it seems that the objective of this article is to conclude that the replicas are better than the originals in terms of resistance to cyclic fatigue.

Author Response

Dear Reviewer 1,

I’m pleased to resubmit the manuscript of the work entitled, “Comparative Study of the SEM Evaluation, EDX Assessment, Morphometric Analysis and Cyclic Fatigue Resistance of Three Novel Brands of NiTi Alloy Endodontic Files

Reviewer 1: his study should only be published as a descriptive analysis of the 3 new file systems, but without comparing them with the previously existing systems on the market.

Response: In order to adapt to the reviewer's 1 comments, we have modified the entire manuscript describing and comparing exclusively the novel brands of Endogal, Path Max Pro and Smarttrack. The study has been focused in the SEM evaluation, EDX assessment, morphometric analysis and cyclic fatigue resistance of the three novel brands of NiTi alloy endodontic files, and the previously existing brands of NiTi alloy endodontic files have been removed from the article. In addition, we have removed all references to the replica- like NiTi alloy endodontic rotary systems, by modifying the aim of the study, following the recommendations of the Reviewer 1. We have also changed all results and Figures.

Reviewer 1: No reference can be made to the fact that they are replicas. That is a lie. None of the new systems has the same alloy, the same taper, the same section, the same pitch, the same helicoidal angle, the same cutting angle.... as the supposed replica!  Figures 2 and 3 itself show that the new brands and the supposed replicas are nothing alike. Figure 5 also demonstrates that the section of ST heat-treated NiTi alloy endodontic reciprocating file is very different to original-brand WOG Gold-wire.

Response: In order to adapt to the reviewer's 1 comments, we have removed all references to the replica- like NiTi alloy endodontic rotary systems, by modifying the aim of the study, following the recommendations of the Reviewer 1.

Reviewer 1: 1) Authors state “In fact, the manufacturers of Endogal, PathMax, and Smarttrack have reported that the geometric design and chemical composition of their instruments are similar and based on those of the ProTaper Next, ProTaper Gold, and Wave One Gold NiTi endodontic rotary instruments, respectively”. However, they do not provide any reference that proves that the manufacturers recognize the copy of other brands. Authors should cite references of manufacturers stating this. It is essential to prove that the manufacturers propose the new systems as copies of the previous ones, which could even bring them legal problems. The title should be modified.

Response: In order to adapt to the reviewer's 1 comments, we have removed all references to the replica- like NiTi alloy endodontic rotary systems by modifying the aim of the study, following the recommendations of the Reviewer 1.

Reviewer 1: 2) In reference to the calculation of the sample size, authors state “We carried out an ANOVA to establish the sample size, achieving 80% power with a confidence level of 5%, with a variability between groups of 0.6 and intra-group variability of 4, in order to identify differences in the contrast of the null hypothesis H₀: μ1 = μ2 102 = μ3 = μ4”. However, they do not specify how many differences they intended to find between the samples. What program have they used? What has been the minimum sample size calculated? With the parameters they give, they would have needed more than 20 files in each group!  The sample is insufficient to validate the results.

Response: In order to adapt to the reviewer's 1 comments, we have rewritten the calculation of the sample size: “A total of 30 experimental units were included, distributed among the 3 groups according to the proportions set by the researcher, and a power of 80% was set, an effect size of 0.606 can be detected when testing the null hypothesis H₀: The means of the 3 groups were equal by means of a 1-Factor ANOVA test for independent samples, taking into account that the level of significance is 5%”. We have modified the sentence in the Material and Methods section.

Reviewer 1: 3) One of the key questions I posed to the authors in my first review was the following:

Authors states “EDG NiTi endodontic rotary files (replica) were compared to PTN NiTi endodontic rotary files (original brand)”. Why do the authors assume that the 25 Endogal file is a replica of the 25 Protaper Next file? Please, include a reference to support this claim! A reference is also necessary to support that Endogal is a Gold NiTi alloy!, How can you compare the cyclic fatigue of a supposedly Gold martensitic alloy file with an austenitic M-Wire file like Protaper Next? Obviously, the result is already known! And the same happens with the rest of the supposed replicas.

Response: In order to adapt to the reviewer's 1 comments, we have removed all references to the replica- like NiTi alloy endodontic rotary systems by modifying the aim of the study, following the recommendations of the Reviewer 1.

Reviewer 1: In their answer, authors state: In order to adapt to the reviewer's comments, we clarify that the manufacturer of the replica-like Endogal NiTi endodontic rotary files ensures that they are replicas of the original-brand Protaper Next NiTi endodontic rotary files. But they do not provide any reference that proves that the manufacturers recognize the copy of other brand. Moreover, the Protaper Next section is completely different from the supposed replica Endogal. Files of different design, different NiTi alloy... cannot be compared to conclude that one has more resistance to cyclic fatigue than the other. It seems that the authors´ objective is to highlight that the replica is better than the original!

Response: In order to adapt to the reviewer's 1 comments, we have removed all references related to the replica-like and original brands NiTi alloy endodontic rotary files by modifying the aim of the study, following the recommendations of the Reviewer 1.

Reviewer 1: 4) Authors state: “In addition, the replica-like Endogal NiTi alloy endodontic rotary files showed a significantly higher time…”. Obviously, since they are comparing a martensitic file (Endogal) with an austenitic file (Protaper Next). I insist, it seems that the objective of this article is to conclude that the replicas are better than the originals in terms of resistance to cyclic fatigue.

Response: In order to adapt to the reviewer's 1 comments, we have removed all references related to the replica-like and original brands NiTi alloy endodontic rotary files by modifying the aim of the study, following the recommendations of the Reviewer 1.

Reviewer 3: Still, I have some minor suggestions: Introduction: Reference no. 15 is pretty old (66 years old). I’m sure that there are newer studies that stated the failures of NiTi alloy can affect the treatment of periapical pathologies

Response: In order to adapt to the reviewer's 3 comments, we have replaced the reference nº 15 by a more recent citation.

We take this opportunity to thank the recommendations and suggestions made by the reviewers to improve the document.

Yours sincerely,

Reviewer 2 Report

The revisions made have addressed my earlier concerns. This has considerably improved the quality of the paper. The authors have documented well in the cover letter and in comments how they addressed each of the points raised.

Author Response

Dear Reviewer 2,

We take this opportunity to thank the recommendations and suggestions made by the reviewer to improve the document.

Yours sincerely,

Reviewer 3 Report

The present study, basically, compare and analyze some properties of replica-like and original-brand endodontic rotary files. There are a lot of such instruments on the market, so, such analyses are welcomed. I appreciate the authors’ work and I congratulate them for their results. In the manuscript that I received, I noticed that some changes were already made, so the quality of the manuscript was improved due to pertinent suggestions.

Still, I have some minor suggestions:

Introduction

  • Reference no. 15 is pretty old (66 years old). I’m sure that there are newer studies that stated the failures of NiTi alloy can affect the treatment of periapical pathologies

Materials and Methods

  • Be sure that the numbering of figures and tables is correct

Results

  • Nothing to mention

Discussion

  • This section starts with “Our conclusions reject…”. I suggest to change the formulation, because if we talk about conclusions, we think about Conclusions section. (it can be changed with “Our results reject…” or something similar)

Conclusions

  • They are supported by the results

Author Response

Dear Reviewer 3,

I’m pleased to resubmit the manuscript of the work entitled, “Comparative Study of the SEM Evaluation, EDX Assessment, Morphometric Analysis and Cyclic Fatigue Resistance of Three Novel Brands of NiTi Alloy Endodontic Files

Reviewer 3: Still, I have some minor suggestions: Introduction: Reference no. 15 is pretty old (66 years old). I’m sure that there are newer studies that stated the failures of NiTi alloy can affect the treatment of periapical pathologies

Response: In order to adapt to the reviewer's 3 comments, we have replaced the reference nº 15 by a more recent citation.

Reviewer 3: Materials and Methods: Be sure that the numbering of figures and tables is correct

Response: In order to adapt to the reviewer's 3 comments, we have revised the numbering of all figures and tables of the manuscript.

Reviewer 3: Discussion: This section starts with “Our conclusions reject…”. I suggest to change the formulation, because if we talk about conclusions, we think about Conclusions section. (it can be changed with “Our results reject…” or something similar)

Response: In order to adapt to the reviewer's 3 comments, we have changed the word.

We take this opportunity to thank the recommendations and suggestions made by the reviewers to improve the document.

Yours sincerely,

Round 2

Reviewer 1 Report

With the proposed new title, i.e. Comparative Study of the SEM Evaluation, EDX Assessment, Morphometric Analysis and Cyclic Fatigue Resistance of Three Novel Brands of NiTi Alloy Endodontic Files, and with the modifications introduced, I consider that the manuscript has been substantially improved and can now be published.

This manuscript is a resubmission of an earlier submission. The following is a list of the peer review reports and author responses from that submission.

Round 1

Reviewer 1 Report

Comparative Study of the Difference in Cyclic Fatigue Re-2 sistance Between Original-Brand and Replica-Like NiTi Endo-3 dontic Files (IJERPH 1530144)

This study aims to compare and analyze the cyclic fatigue resistance of replica-like and original-brand endodontic rotary files made from nickel-titanium (NiTi).

This article has much merit, however, some serious methodological errors make the study results invalid and prevent its publication. Moreover, the Introduction needs a lot of clarification and references. The authors should reconsider the objectives of the study. They should consider including the results that they present in Mat and Met among the objectives, or else eliminate them and prepare a new manuscript with those results. The manuscript must be completely rewritten. The continuous mention of replicas and originals must be justified with references or eliminated.

Introduction:

Although Introduction carry out a correct review of the necessary antecedents to justify the objectives of the study, the authors make some statements that need references to support them.

The statement “More recently, novel brands of NiTi endodontic rotary instruments have emerged to enhance mechanical properties and reduce rates of instrument failure; however, the geometrical design and metallurgical properties of some of these novel NiTi endodontic rotary files resemble and appear to be based on previously manufactured instrumentation systems” should be clarified.

What new brands of NiTi endodontic rotary instruments are the authors referring to? This is an essential point of the Introduction. Are Endogal, PathMax and Smarttrack files included here? If so, they should reference and explain what type of alloys these files have, what mechanical properties and what other characteristics they have.

Why are they based on previously manufactured instrumentation systems? Why are they replica-like brands? Does that imply that they are copies? Have any rights or patents been infringed? These statements needs explanation, and references.

Is the metallurgical composition of these files known? This is a fundamental point. How can it be said that they are replicas if the composition of the NiTi alloy with which they are made is not known? Cyclic fatigue resistance is highly dependent on the type of NiTi alloy the files are made of.

The authors should modify the objective of the study. In this study, 6 types of files will be compared, but it cannot be assumed that some are replicas of others, since the composition of the NiTi alloys with which they are manufactured is not known. No study that concluded that some systems are replicas of others is cited.

Do the authors assume that the 25 Endogal file is a replica of the 25 Protaper Next file? This should be clarified.

Material and Methods:

How was the sample size calculated? What program was used?  Are 10 files enough for the results to be meaningful? Are 10 files enough for the results to be meaningful? What program was used? what statistical significance was established? what power did you set? what difference was it intended to detect?

Authors states “EDG NiTi endodontic rotary files (replica) were compared to PTN NiTi endodontic rotary files 100 (original brand)”. Why do the authors assume that the 25 Endogal file is a replica of the 25 Protaper Next file? Please, include a reference to support this claim! A reference is also necessary to support that Endogal is a Gold NiTi alloy!, How can you compare the cyclic fatigue of a supposedly Gold martensitic alloy file with an austenitic M-Wire file like Protaper Next? Obviously, the result is already known! And the same happens with the rest of the supposed replicas.

The statements “PMP NiTi endodontic rotary files (replica) were compared to PTG NiTi 101 endodontic rotary files (original brand)” need reference. PMP is heat-treated while PTG is Gold! Why do you say that it is a replica and its original? You need a reference to support it.

The statement ST NiTi endodontic reciprocating files (replica) were compared with WOG NiTi endodontic reciprocating files (original brand), also need reference. ST is heat-treated while WOG is Gold! Why do you say that it is a replica and its original? You need a reference to support it.

What does it mean that the replica selection of the original-brand NiTi endodontic rotary systems was carried out in accordance with the manufacturers' instructions?

The expression "original-brand and replica-like" should be omitted throughout the manuscript unless references are provided to support this statement. The pairs of files being compared are not only made of different NiTi alloys, but also have a different structure, as shown in figures 1, 3 and 4. Moreover, Table 1 shows that the atomic weight percent (%) are quite different between them, and table 2 shows quite different pitch measurements (mm). Why are they replicas and originals?

The authors include in the Mat & Met section a series of results that should not appear in this section: Analysis with Scanning Electron Microscopy, Energy-Dispersive X-Ray Spectroscopy, Analysis with Micro-Computed Tomography Scan and Morphometric.

If the aim of the study is to compare resistance to cyclic fatigue, the authors should state in Mat & Met only how resistance to cyclic fatigue was determined. If they want to include these results, they should move them to the Results section and broaden the objective of the study.

Results and conclusion

Logically the results show something that is already known: that files made with austenitic alloys, such as the M-wire with which PTN is manufactured, have lower resistance to cyclic fatigue fracture. The results also show something well known: that heat-treated NiTi alloys (PMP and ST) also improve resistance to cyclic fatigue fracture, compared to Gold alloys. These results are widely known to all endodontists and do not add anything new.

Discussion

In the discussion, results presented in Mat and Met are discussed. This needs to be changed.

The composition of the NiTi alloys was not an objective of the study. Why is it included in Mat & Met and why is it discussed?

The main objective of the study, cyclical fatigue, and its results, should be better analyzed in the discussion. Moreover, the results should be better compared with the results of other studies.

Conclusion

The conclusion, The present study concluded that replica-like NiTi endodontic files have higher cyclic fatigue resistance than original-brand NiTi endodontic files, should be re-written without reference to replica and original.

Reviewer 2 Report

I think the author should update the references in the introduction sections by adding this document:
Evaluation of pressure distribution against root canal walls of NiTi rotary instruments by finite element analysis. Applied sciences 2020, 10(8), 2981

- The influence of brushing movement on geometrical shaping outcomes: A micro-CT study. Applied sciences 2020, 10(14), 4805

  • Micro-CT evaluation of rotary and reciprocating glide path and shaping systems outcomes in maxillary molar curved canals. Odontology 2021 I think the paper is well written, the method are appropriated. The study offers a different point of view about instruments brand comparing them with the gold standard one known by the clinician, this in my opinion it could help to approach those instruments also by generalist dentist. The author may think to explore, as the already had the cad instrument, a fem simulation for the point of pressure against the canals walls.
I think the paper is well written, the method are appropriated. The study offers a different point of view about instruments brand comparing them with the gold standard one known by the clinician, this in my opinion it could help to approach those instruments also by generalist dentist. The author may think to explore, as the already had the cad instrument, a fem simulation for the point of pressure against the canals walls.

Reviewer 3 Report

The paper needs the following editing changes made.

The introduction should mention the influence of chemical factors in the degradation of NiTi files e.g. NaOCl as discussed in these papers
https://pubmed.ncbi.nlm.nih.gov/19436252/
https://pubmed.ncbi.nlm.nih.gov/17209831/
doi: 10.1111/j.1365-2591.2006.01178.x.
and that corrosive effects are time dependant

Line 65 There are now lab models such as 3D printed teeth that can overcome variability due to anatomy 
https://pubmed.ncbi.nlm.nih.gov/34967333/
and this should be mentioned either here or in the discussion (e.g. line 375).

Line 83 Specify whether the files were supplied sterile by the manufacturer

Line 86 Missing space between the words Spain and NiTi

Lines 94 and 96 The batch number is missing for F2 ProTaper 94 Gold and Smarttrack

Include scale bars in each panel of Figure 1.

Each panel of Figure 1 needs adjustment for grey scale levels (gamma and contrast).

For Figure 2, take out the miniature tables as those are too small to read. There is no need to have these as that data for atomic composition in percentage for the 6 samples for the key elements is already shown in Table 1.

For Fig 4, rearrange the separate pieces into a grid of 2 columns X 3 rows (with relevant pairing of original and replicas) showing the cross sectional profiles at much greater magnification as this is a very important characteristic.  

Line 168 Researches should be Researchers

Line 177 Provide an image or schematic of the artificial root canal

Line 191-208 To prevent repetition and increase transparency, replace this paragraph with a table stating the sample type and the test parameters

Line 218 Explain how data sets were checked for normality prior to parametric statistical analysis

Line 230 incomplete word "intensity and precision of this aser machining process"

Line 237 A 20 kV acceleration voltage will give deeper penetration into the surface of the sample and a higher signal to noise ratio than 15 kV. This is a better explanation than the words "a more in-depth analysis of the surface"

Table 2 is not needed as it just repeats data that is already shown in Figure 3.

Line 279 This should read   ...the Tukey post-hoc tests also found....

Fig 6 is not needed as it repeats what is shown in Table 4.

Table 5 has a formatting error in the first column.

The number of decimal points in the data shown in Table 5 should be 1 rather than 4.

Line 382 Referencing error [10,Error! Bookmark not defined.]

Line 396 This should read: Dynamic cyclic fatigue testing 

Style errors are present in the references: 11, 13, 15, 23, 25, 29 (incorrect use of capitals in the title)